# Recovery-Oriented Practices in a Mental Health Centre for Citizens Experiencing Serious Mental Issues and Substance Use: As Perceived by Healthcare Professionals

**DOI:** 10.3390/ijerph191610294

**Published:** 2022-08-18

**Authors:** Kim Jørgensen, Morten Hansen, Bengt Karlsson

**Affiliations:** 1Science in Nursing Department of Public Health, Bartholins Allé 2, 8000 Aarhus, Denmark; 2Psychiatric Outpatient Clinic Ishøj, Bostedsteamet, Store Torv 20, 2635 Ishøj, Denmark; 3Center for Mental Health and Substance Abuse, Department of Health, Social and Welfare Studies, Faculty of Health and Social Sciences, University of Southeastern Norway, Postbox 7053, 3007 Drammen, Norway

**Keywords:** recovery, hope, relationships, connectedness, user involvement, mental health services, inpatients, person-centred care

## Abstract

Introduction: Recovery-oriented practices have become a means of promoting user recovery during hospitalisation, but we do not know much about the concrete means of practicing recovery-orientation for the most vulnerable users with serious mental difficulty and substance use. Aims: We investigated the concrete means of practicing recovery-orientation in care work and the elements, dimensions, outcomes, or steps of it in a special department of mental health centres. Method: Focus group interviews were conducted with 16 health professionals with experience with users with serious mental difficulty and substance use. Qualitative content analysis was undertaken. Results: The main theme was “holistic recovery on structural terms” based on two themes and four subthemes. The first theme was “recovery based on an individual approach” with subthemes “detective—find hope” and “how to do recovery-oriented practice”. The next theme was “recovery subject to structural framework” with subthemes “tension between different interests” and “symptoms as a barrier”. Conclusions: recovery-oriented practice is understood as an approach where health professionals emphasise forming relationships based on trust, being hopeful for the users’ future, spending time with users, and respecting users’ experiences and knowledge from their own life. There are cross-pressures between different interests. The desire to meet the users’ perspectives and respect these perspectives but at the same time live up to mental health centre purposes to stabilise the users’ health and achieve self-care.

## 1. Introduction

Danish health policy documents set objectives for a more holistic approach in the mental health sector with initiatives and goals. One of the political goals is deinstitutionalisation and a focus on recovery and opportunities to live a good life with one’s illness or to recover completely [1,2].

In terms of health policy, a mental health practice is desired in which the users must be equal and active partners in their course of treatment [3,4,5]. A recovery-oriented approach is used in Danish mental health centres [6]. In many Western countries, recovery has become a political goal for mental health practices [7,8,9,10,11,12]. Several researchers refer to the development as a paradigm shift from a traditional biomedical expert-dominated approach to a humanistic, holistic approach, where the focus is on the individual’s well-being, resources, and desires to achieve a meaningful and hopeful life, where the individual’s experiential knowledge is included [13,14,15].

However, research into recovery is in its infancy, and knowledge about how the concept unfolds in mental health centres, where the target group is the users with the most serious disorders, many of whom also use substances, is lacking. A large part of the research is more about finding methods such as cognitive behavioural therapy or organising bids as teams and testing them as efforts for involvement and recovery without defining the concepts theoretically or empirically [16,17].

It may be helpful to look at the connectedness of different actors to the idea of user-centred care by understanding how recovery has become such a strong ideal in healthcare. Strong user movements in mental healthcare strongly support a more recovery-oriented direction (user movements in Denmark include The Social Network, The National Association for Mental Health, The National Association Of Current And Former Psychiatry Users (LAP) And Better Psychiatry—The National Association For Relatives). The user movements draw attention to the fact that recovery constitutes civil law requirements, which means equal access to mental health service rights to participate in planning the course of mental health [18,19]. The practitioners of mental health services are accused of being too paternalistic in decision-making processes and of being preoccupied with diagnoses, symptoms, and medication [20]. Furthermore, in the last decade, there has been criticism of the use of coercion, and, from a political point of view, aims and efforts have been developed to reduce this. A recovery-oriented focus is projected in this regard as the key to avoiding coercion [21,22]. The literature shows that “recovery” has clues to many similar concepts, such as “participation”, “influence”, “empowerment”, “inclusion”, “peer support”, “co-creation”, “compliance”, “self-efficacy”, and “health literacy” [8,13,23,24,25] in combination with terms such as “patients”, “citizens”, “clients”, and “service users” [26]. However, the existing research literature is limited concerned with how recovery-oriented practice unfolds at the moment in a mental health centre [27].

Users often find it difficult to move forward in their course, for example, after a discharge, and there is rarely a professional who follows-up on the users and helps to establish and coordinate the efforts in a mental health centre or municipalities [28,29]. Previous research has focused on investigating one sector or on the actual transition between the sectors and not across lengthy courses, which often include mental health centres and municipal rehabilitative follow-up [30,31,32,33].

Research shows that the users’ course is mainly framed and structured based on the understanding of ‘clinical recovery’, i.e., with a focus on treating diseases and symptoms medically. For many users, this viewpoint means that their needs are not met based on a holistic understanding, nor do they receive the necessary psychological and social help during their course. The consequence is that many users are often readmitted and do not start a satisfactory and meaningful social life [34].

Mental healthcare in hospitals contains many specialisations, mainly based on diagnostic criteria. In this study, the focus is not on diagnoses, but “the special places” in mental health centres, which are aimed at all kinds of diagnoses and various psychological and social difficulties. However, a large group of vulnerable users have severe mental disorders, some with violent behaviour that exhibit unpredictable behaviour, have repeated hospitalizations, interrupt treatment courses, and often use drugs and/or are required to undergo treatment. Some of these users have been violent towards the professionals in social psychiatric accommodation, in some cases resulting in death.

On that basis, the Danish government has adopted an amendment to the Psychiatry Act [35] to set up a special mental health centre department to help this target group of users with severe mental illness and substance use. These mental health centres in hospitals are referred to as “the special places”, where the users are admitted from six months to several years. The overall political goal is through a recovery-oriented approach and further follow-up in the municipality to prevent the users from being involved in conflicts and episodes of violence that create insecurity among other residents and employees. A hospital stay from six months to several years is significantly longer than the average time. A recovery-oriented approach can enable users to live an independent and satisfying life, overcoming difficult mental and social difficulties [35,36]. The recovery-oriented efforts in the special department of mental health centres include both a social work efforts and voluntary substance use treatment, as well as daily activities, such as exercise, creative outlets, and the goal of improving a patient’s ability to master everyday life [37] in addition to ordinary mental health service. Recovery-oriented practice must strengthen users’ empowerment [38] and support socialisation [39] to encourage the independent seeking of work, wellness, and interpersonal relationships. We have limited knowledge of how recovery-oriented practice is integrated into inpatient settings [40,41,42]. As it appears, the political and administrative point of view intends for the special department of mental health centre to work as a recovery-oriented department to improve users’ lives and options for action. However, we do not know what this entails. We wish to generate knowledge by creating learning and optimising treatment and rehabilitation. The aim is to investigate the concrete means of practicing recovery-orientation in care work and the elements, dimensions, outcomes, or steps of it in the special department of mental health centres.

## 2. Method

The study is based on a social constructivist framework where individuals are active participants in the creation of their own knowledge. The theory of social constructivism focuses strongly on dyads and small groups [43,44]. Through social interactions, participants’ experiences and insights about recovery-oriented practice become clearer. We have designed an explorative research project to identify, describe, and transfer knowledge about health professionals’ experiences with a recovery-oriented practice [44].

### 2.1. Sampling

We used purposive sampling [45] to ensure the recruitment of participants who could provide in-depth and detailed information about the phenomenon under investigation. The first author contacted the management at a mental health centre. There was great interest in participating, and the management communicated the contact to the participants and agreed on which days the interviews could take place. The first author contacted the participants. The first author informed them about the purpose of the project and their legal and ethical rights as participants and obtained their informed consent.

In total, 16 participants gave their informed consent to participate, many of whom work with users in the special department for users with serious mental difficulties and many with substance use problems. Most healthcare professionals had 2–4 years of experience in the department. A few healthcare professionals had significant experience in other fields of mental health and had been employed for more than seven years. The participants were mostly of a mature age and worked as nurses, social- and healthcare workers, physiotherapist, pedagogues, and unskilled workers. The research group consisted of three professionals. Two researchers are trained mental health nurses with PhDs and an educator employed as a peer mentor with user experience of mental health services.

### 2.2. Focus Group

In order to obtain nuanced perspectives on health professionals’ experiences with a recovery-oriented practice, focus group interviews were chosen. A key strength of focus group interviews is the ability to obtain rich data by bringing together a group of 4–10 health professionals who work together and facilitating a discussion of their experiences of working with recovery-oriented goals [46]. All interviews were audio-recorded and verbatim transcribed. Open, reflective and delimiting questions were used to open up a discussion while ensuring an elucidation of the study’s specific objectives [47]. The interviews were held at the mental health centre to promote confidentiality among the participants. Inspired by existing knowledge in the field [33,48], we formulated a list of questions to explore our participants’ experiences with a recovery-oriented practice in mental healthcare. Based on the questions, we asked a few broad questions [49] and focused on follow-up questions to achieve wealth and depth (Table 1).

The development of themes provided a framework for the knowledge we wanted to explore. However, in the focus group interviews, we let ourselves flow with the thoughts the participants had expressed. We asked more deeply about these thoughts to achieve a nuanced understanding of their opinions [47,50,51].

### 2.3. Ethical Considerations

The health professionals were included and gave their informed consent to participate after receiving oral and written information about the purpose of the project and the legal and ethical rules for participation. We adhered closely to the ethics for scientific work. The study was accepted by the Danish Capital Region Data Protection Agency (j.nr.:P-2020-345) and was carried out according to the Helsinki Declaration [52] and Danish law [53]; no formal permit from a biomedical ethics committee was required, as the purpose of the research was not to influence the informants, physically or psychologically.

### 2.4. Data Analysis

To ensure the validity of this study, we carefully selected rigorous data material and data saturation and systematically treated the material through in-depth exploration using qualitative content analysis [54]. We use manifest content analysis to carry out a low-level analysis of the participants’ experiences in mental healthcare. The systematic analysis process consisted of four stages: condensed meanings, categories, subthemes and themes. In the first phase, we read the transcribed interviews across the sample and identification of meaning units. We identified each concept and divided it into meaningful units, which we subsequently coded. The code sheet was developed in an iterative process. The transcribed interviews were initially coded as having relevance in relation to the aim of the study. We developed a preliminary code sheet that was subsequently revised. Along with the analysis of the texts, the code sheet was developed as new codes appeared. The codes were discussed by the research team and the codes were categorized into sub-, theme-, and main-themes. The developed themes formed the basis for a coherent understanding and were viewed concerning the aim of the study. Throughout the process, we repeatedly returned to the original text to compare and ensure our understanding [54]. The content analysis was supported by the NVivo software [55].

The themes reflected a new understanding of the concrete means of practicing recovery-orientation in care work and the elements, dimensions, outcomes, or steps of it in a special department of mental health centres.

## 3. Results

The focus group interviews were facilitated through the interview questions, and according to the social constructivism approach all participants were active and contributed to listening to each other and the understanding and opinions developed during the interviews.

The main theme, “holistic recovery on structural terms”, was based on two themes and four subthemes. The first theme was “recovery based on an individual approach” with subthemes “detective—find hope” and “how to do recovery-oriented practice”. The next theme was “recovery subject to structural framework” with subthemes “tension between different interests” and “symptoms as a barrier” (Table 2).

### 3.1. Recovery Based on an Individual Approach

Recovery was understood as a recovery-oriented individual practice, which means meeting the users’ needs and hopes and helping them realise this truth in their lives. In the discussion, most participants are preoccupied with finding a way to meet the individual in his or her hopes and aims for the future. Under the above theme, two subthemes were discussed: “detective—find hope” and “how to do to recovery-oriented practice”.

#### 3.1.1. Detective—Find Hope

All the participants strive to see the users as individuals and do not find diagnoses and substance use interesting.

*‘We do not go so much into what diagnosis they come up with, but more what their backgrounds and problems are right now and here. Many of them have been abused and have themselves been abused by others. They know psychiatry inside and out and have all been subjected to coercion to a greater or lesser degree. Most people have or have had an addiction, and therefore often have some form of abusive behaviour’* (FG (FG Focus group interview) 1).

In the discussion it became clear that the participants often think differently and were inspired by each other. Most participants think the most important objective is to find out what is important for the users and what hopes and goals users have for the future. For many users, mental health symptoms, e.g., such as hallucinations, are not the worst. For some, it is more important to make a good friend or reconnect with their family, the dream of getting a new home, an education, and a job, being able to self-care without professional help, less substance use, etc. Drug and alcohol abuse is used for self-medication and its use leads to social conflicts. Some users are abused by other users in order to obtain drugs or to abuse themselves.

*‘So, I think recovery means a way to find a new meaning and purpose in life and live a good life despite symptoms’* (FG 2).

The core assignment is to understand what the individual user finds important in his or her life and to be asked open-ended curious questions that talk more about the individual’s values. Some users have a harder time expressing themselves linguistically. Here it can help express oneself through social contexts or an activity. It could be a social activity that promotes users’ joy by entering relationships by doing something with others, e.g., cooking together.

*’It is not certain that the individual user perceptions of what future life should be like are the same as that of us employees. So that’s what you must find out. What does the individual user consider to be a good life? This is what recovery-oriented work is about. You must find out and work towards’* (FG 1).

*‘e.g., tells a user that she has not seen her aunt in five years. Now I really want to get in touch with her. Then that’s what we’re working on. Then that’s what’s most important. Any user needs something more visual to find their hopes. e.g., we make some cards that say “family” or “personal”. Then you can talk from it and find out what it is that is important’* (FG 2).

Another important aspect that several participants mention is being patient in listening to users’ needs, while being careful not to push professional aims and ambitions on users, in a way that users’ recovery is not the focus but rather the wishes of the participants regarding the lives of the users.

*‘We want our users much, but sometimes we get to set expectations too high in relation to what it really is they want. So, what about, just finding out what their wishes for the future are, and not what we would like because we just want the very best for them’* (FG 1).

Some participants consider the users’ history an essential basic subject that reveals some aspects of who they are. It can, e.g., tell something about their positive/negative experiences, relationships with other people, interests, and what their medical history has been.

*‘We are completely out to find out how the pregnancy has been, how the upbringing and school conditions have been. This is how we form this whole story about this human being. Then we find out how their brain works and how we can work with them. Then the long phase begins, where we start with different things’* (FG 1).

Their life story thus becomes a point of reference and starting point for how professionals attempt to understand users’ needs. It is a point of view that becomes powerful in regard to saying something about the future. The other informants do not contradict this approach, but one could take the opposite view and believe that the past is not necessarily crucial for current and future perspectives.

#### 3.1.2. How to Carry out Recovery-Oriented Practice

All participants considered a good and close relationship between professional and user as a prerequisite for good recovery-oriented practice. A confident and close relationship is perceived as a means of gaining openness and insight into the user’s life situation.

*‘The relationship plays a big role. When the relationship is established, the user may come and say I have a bad day today, or I am frustrated, or something. When you know each other well, we know what has worked in the past. e.g., do you want to go to the gym, or should we go down to the workshop and repair a bike. Or what is it you need right now? Then we talk about what it is that works’* (FG 2).

The participants consider their relationship with the user as the most important tool to be able to help. It is about empathy and sympathy for the other to understand how the user feels and what he or she needs.

*‘We come with our differences, you could say. We can bid with our knowledge, our educations and even as individual’* (FG 1).

Recovery-oriented practice is linked to a time perspective, as sufficient time is needed to help. It is emphasised that most users are hospitalised for at least one year, while several are hospitalised for numerous years. The target group of users has many hospitalisations behind them, severe mental difficulties, and a history of substance use. The long-term perspective provides great opportunities to build confident relationships and offer individual help and support to promote user recovery.

*‘We have some nice advantages here compared to other places in psychiatry, where people are in short courses. Our users have been here for a long time’* (FG 1).

It takes time to figure out what the users want from admission. Experience shows that one cannot expect users to answer what they want to use the hospital for at the start. What you want with your life or what you dream about are big questions, and they need understand and find themselves before they can answer such important questions. Several participants wish to create an environment with structure and professional knowledge, where there is peace so that patients can thrive and have the opportunity to develop and achieve recovery.

*‘Our patients need to learn to get along with each other and feeling secure. Many of them may have several years in psychiatry. So, it is also a learning to come here in a social community with others and where there is an expectation of participation in an active everydaylife’* (FG 2).

When participants talk about how to carry out recovery-oriented practice, they find it important to involve users’ resources, involve them in decisions, listen to their life experiences, and respect their knowledge and wishes.

*‘We work with the four corners. It means we do things together with the users because it’s something all people need to feel included. Also, to be able to use it as one can. We are not wiser than users, but we listen and learn together and meet users where they are and try to take their life experiences in, and together, we become people. Working this way is like growing a tree in the middle’* (FG 1).

### 3.2. Recovery Subject to Structural Framework

The structural framework conditions challenge participants’ holistic approach to helping users in a recovery-oriented direction. Health policy and the economy play a significant role in what admission is aimed at, and here expectations are set that the user will, as far as possible, manage on their own, independent of professional help, after an admission. In addition, it turns out that users’ difficulties due to their mental illnesses and substance uses can also make it difficult to work in a recovery-oriented direction. Under the above theme, two subthemes were discussed: “tension between different interests” and “symptoms as a barrier”.

#### 3.2.1. The Tensions between Different Interests

The target group of users for the special department for mental health are visited in a collaboration between the municipality they live in and the inpatient mental health centre. The municipality must pay for most of the hospitalisation, and therefore only those users who have difficulty living their everyday lives are examined. Users often have many admissions behind them and, for the most part, partake in significant drug use.

*‘The user has been lost in many places. Some of them are not even allowed to come to the municipal office because they are threatening and such. So, the thing about coming to an addiction centre in a municipality and showing up and being clean, neat, and freshly washed and having to sit down around a table, they cannot’* (FG 2).

During hospitalisation, the purpose of a user’s hospitalisation is also stated. The paying home municipality participates in the preparation of a plan together with the professionals at the mental health centre and the user. Here, a plan is developed to ensure a path towards a more independent life and, if possible, where users can live in an independent home, possibly in a shared residence with others. Efforts must be made to move the user out or reduce substance abuse and to overcome the mental difficulties that come with diagnoses.

*‘We now demand that we are in contact with the municipality at least every three months because we must always have the municipality involved in what it is we must achieve during the hospitalisation, what users want, and what the municipality wants. The day we discharge from here, the municipality must have an offer because otherwise, they will be bad at going and waiting for nothing. Then we can start over. It is also important for the municipalities to understand whom they are visiting for admission. So, they do not just use it as: Well, yes, they are making trouble at the residences. We cannot place them. Then they just come to the special places’* (FG 1).

Several participants state the municipality’s requirement that users must be admitted to the special places in psychiatry in order to achieve a better and more independent life. Furthermore, the task is to reduce the users’ drug use and possibly violent behaviour, i.e., reach a group with many hospitalizations behind them in traditional psychiatry.

#### 3.2.2. Symptoms as Barrier

Participants were concerned with offering a recovery-oriented practice and involving the user perspective to the maximum. The path to recovery is based on users’ active participation in activities and motivation, e.g., end their substance use. Severe mental and social difficulties challenge the collaboration between users and health professionals.

*‘We have had many with severe symptoms and abuse problems. Common to them all is that it can be difficult to rehabilitate and work with someone who is highly paranoid and has a major addiction. So, it is clearly our goal to reduce the abuse or stop it altogether’* (FG 2).

Several participants are especially frustrated by the massive use of drugs or alcohol among users as it exacerbates the recovery-oriented work. It goes beyond their concentration and motivation, and they become tired or euphoric.

*‘It can be said that, in principle, it is not allowed to abuse here. It’s a hospital, and we cannot witness abuse here, but there are not the big consequences if one has abused and fallen into, one might say’* (FG 1).

When a health professional suspects a user has taken drugs on having left the department, the staff can ask for a urine test to ascertain whether drugs have been taken. If there are substances in the urine, it gives rise to a conversation with the user. If the user is not motivated to stop the substance use, there is not much more they can do. This situation is highlighted as a problem among the participants because it prevents them from doing a good job.

*‘We can actually forcibly detain them because we work under the Psychiatric Act, but the substance use in itself is not enough to detain users. Instead, we try to make agreements we make with them because they really want to help themselves’* (FG 1).

Several participants experienced that the relationship of trust is challenged when, on the one hand, they must be controlling regarding substance use and, on the other hand, be listening and accommodating to the users’ own wishes.

*‘I feel both I am a guardian while I try to show confidence that I believe in users’* (FG 1).

## 4. Discussion

The results from this qualitative study show how health professionals understand and work in a recovery-oriented manner with a target group of users that differs from traditional mental health centre departments and here with a hospital admission that extends from half a year up to several years for some. Despite the special department in mental health centre being specifically aimed at users with serious mental health and abuse problems, it seems the result is similar to other studies in mental health centres. Similar to other studies, recovery-oriented practice addresses the importance of relationships, trust, and being hopeful [29,56,57].

The analysis of the data shows that the participants do not clearly define recovery, but the participant understanding of recovery can be reflected in Anthony Williams’s definition of personal recovery as “a deeply personal, unique process of changing one’s attitudes, values, feelings, goals, skills, and/or roles”. Recovery involves the development of new meaning and purpose in one’s life as one grows beyond the catastrophic effects of mental illness [58]. As other studies show, methods to promote a recovery-oriented approach are often mentioned, e.g., cognitive behavioural therapy, psychoeducation, and environmental therapy [3,27,56,57,58], but in this study, the participants perceive relationship work as essential for recovery. A relationship was understood as a confident and acknowledged close relationship where both individuals could express thoughts freely and be met with respect and curiosity. In other studies of relationships, emphasis is placed on the prerequisite for establishing a fruitful collaboration, where the user shows trust and openness and where a good relationship can arouse motivation for change in users [10,11,24]. Like other mental health centres in Western countries, structural conditions, efficiency, and neoliberal societal development mean that a recovery-oriented practice that aims for the maximum involvement of users’ perspectives has narrow conditions [3,13,27,59,60]. The target group of users in this study have their hospitalisation financed by their home municipality. The municipality demands that during hospitalisation, work is carried out in a positive direction regarding users’ behaviour so that it leads to a positive change. It involves no or less violent behaviour, less or no substance use, more self-care, strengthened empowerment to deal with one’s own problems, and, if possible, moving into an independent home, and motivation for education and jobs are also on the municipality’s wish list [35]. Thus, the result shows that the special places are subject to the same structural conditions that we know from traditional mental healthcare in Western countries [17,19,61,62]. The outcomes are an approach that emphasises clinical recovery, where the focus is on becoming symptom-free, achieving a better level of function, achieving self-care, and becoming a contributing citizen in society [56,57,59].

Recovery-oriented practice occurs in a cross-pressure between different humanistic and structural values, which in turn is influenced by political power relations and societal ideals. Regardless of good intentions, health professionals can meet the individual user’s wishes and hopes. The recovery-oriented approach is subject to the strategic objectives that apply in a hospital system with the task of diagnosing and treating people with a disorder [4,27,59,60]. Another consequence could be, as other studies show, that the health professionals use a paternalistic approach, which has been a barrier to user involvement and personal recovery [24,60]. Research shows that many users are readmitted because it is difficult to find meaning and deal with psychosocial challenges after discharge [28,62]. This study differs from other studies by showing how professionals commit to meeting vulnerable users’ needs, but, regardless, are subject to societal norms and neoliberal development trends.

## 5. Limitations

The data consist of two focus group interviews with sixteen participants with different educations, e.g., nurses, assistant nurses, educators, physiotherapists, and occupational therapists. In some of these, we did not obtain our desired number of twenty participants. Thus, we chose to conduct two focus groups with sixteen attendees, one with seven and one with nine. The groups with seven participants did not achieve the same dynamic conversation that typically characterises focus group interviews. However, despite the few groups, these groups ended up providing rich answers that helped form the basis of the analysis.

## 6. Conclusions

The aim was to investigate the concrete means of practicing recovery-orientation in care work and the elements, dimensions, outcomes, or steps of it in a special department in mental health centres. The participants do not work with a specific theoretical approach or method but base their responses on the experiences of what works well for users. There is no clear definition of recovery, but their answers are related to the definition of personal recovery of [58]. The participants are deeply involved in aiding users and especially highlight the work of relationships as an essential approach. Trust, interest, spending time with users, and being hopeful are some of the values the participants attach great importance to in their recovery-oriented work.

On the other hand, there may be cross-pressures between different interests. The purpose of legislation regarding special departments was to stabilise users’ health and improve their ability to cope with everyday life, reduce the number of episodes of violence and conflicts, and prevent coercion against the target group. Thus, for example, the goals challenge different interests. There is an expectation of positive change if the change is defined by the system, which does not necessarily harmonise with the users’ wishes for recovery or the health professionals’ efforts to meet the users’ hopes and dreams for the future. For example, the user may wish to continue his or her substance use while the health professionals have the task of removing or reducing it.

## Figures and Tables

**Table 1 ijerph-19-10294-t001:** Interview guide and data gathering.

Thema	Research Questions	Interview Questions
Recovery	How is the concept of recovery experienced by healthcare professionals?	- What does recovery mean for you as a health professional? (there are several perceptions of recovery, so the self-perception is the basis for the understanding of answers in the next theme) - Can recovery be achieved? - Do you have any criticism of the phenomenon of recovery/is recovery limited?
Recovery-oriented practice	How is the health care effort made recovery-oriented for the patient? How do healthcare professionals work in a recovery-oriented manner?	- What does it mean for you to work in a recovery-oriented manner? - In what ways do you feel that you work in a recovery-oriented manner? - Does the recovery-oriented approach have limitations? What works/does not work?
Structure	How are health professionals and recovery-oriented efforts structured in general? How is the process structured for a patient?	- Can you describe a patient course and how this is structured? - How is recovery considered in the structure of treatment? - How is everyday life in the ward structured for you and your patients?
Meaning/hope/goal	How are meaning, goals and hopes perceived, and what significance does it have for the treatment?	- What does hope mean to you, and what does it mean for the patient’s process? - What does it mean for the patient to find meaning in the process and their situation? - How do you help the patient set goals and ensure that the patient is motivated to achieve these?
Examples of recovery-oriented practice		- Can you come up with concrete examples of how you work in a recovery-oriented manner?

**Table 2 ijerph-19-10294-t002:** How health professionals experience recovery-oriented practice in the special department of a mental health centre: subthemes, themes, and main theme.

Subthemes	Themes	Main Theme
Detective—find hope How to do recovery-oriented practice The tension between different interests Symptoms as a barrier	Recovery based on an individual approach Recovery is subject to a structural framework	Holistic recovery on structural terms

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
