# Peer review of "Recovery-Oriented Practices in a Mental Health Centre for Citizens Experiencing Serious Mental Issues and Substance Use: As Perceived by Healthcare Professionals"

_ijerph, 2022, doi:10.3390/ijerph191610294_

Round 1

Reviewer 1 Report

Dear authors,

Thank you for the opportunity to review your work. As it stands, the manuscript I very well written and deals with an important issue.

I have provided some suggestions that could help improve the manuscript e.g. by pointing to some aspects that are unclear to an outside reader. 

Abstract

In the sentence “but we do not know much about how research works with recovery-13 oriented care for the most vulnerable users with serious mental difficulty and substance use” it is not clear what is meant by “how research works with…”

Introduction

P. 1, line 32: what is meant by ‘a more humane approach’?

P. 1 line 35-36: I suggest rephrasing this sentence – perhaps move to method section and provide more context about the study setting. (for example, with some of the information provided in p. 2. Line 77-88)

P. 1, line 37: delete ‘also’

P. 1, line 42-44: I suggest rephrasing as follows: “… in its infancy, and knowledge abouthow the concept unfolds in mental health centres, where the target group is the users with the most serious disorders, many of whom also have substance use is lacking

P 2, line 45: please clarify what methods the text is referring to.

P 2, line 67-68: I suggest rephrasing the first part of the sentence – e.g. Previous research has focused on…

P. 2, line 77-82: This section seems to lack coherence – I suggest rephrasing.

Using the term ‘the special places’ makes good sense to me, as I am familiar with the Danish mental health context, however it might be difficult to follow to other readers, who are not. It could be helpful to make it very explicit, that this is actually the term used about the study setting. Perhaps simply by marking it with ‘…’

P. 3, line 102: replace create with generate knowledge 

Method 

This section is we written and works well to present the interview topic/question guide. I suggest providing some more demographic information about the participants such as occupation (nurses, social- and healthcare workers…), age (range or mean) and gender. Some is mentioned in the discussion/limitations, however I believe some more detailed information about this is important for readers to be able to understand the results, the quotes used etc. 

Furthermore, I suggest restructuring the section as follows – and consider changing some of the headings:

Sampling (and participants)

Focus groups

Ethical considerations

(Data) Analysis

The section on ethical considerations is repeated at the end of the manuscript – however with a different content. I suggest collating the two sections.

Result

The results are very interesting and I get the points made. However, it would be helpful if the authors could provide some context related to the presented quotes. Also, when thinking about the study being placed within a social constructivism perspective. This could be done by providing some context about when and how the quote appeared in the focus group discussions.

Furthermore, I suggest being more explicit about the points presented as coming from the participants. As is stands, the text is not clear about this, and the lack of contextualisation around the used quotes breaks up the flow of the text.

I suggest providing a short sum-up-point at the end of the result section.

Some other specific suggestions:

The first sentence is unnecessary and can be omitted.

Table 2: This is perhaps a personal preference however, I would turn the table around and present the main theme then the themes and subthemes. 

P. 6, line 191: combine the sentences in line 191 and 192 with ‘and’.

Discussion

P. 9, line 372-375: the end of this quote is not clearly marked and please add specific page number to where the quote is from.

P. 9, line 387: ‘commune’ is this the right word? I suggest replacing it with municipality.

Conflicts of interest: Two of the authors could consider declaring that they represent guest editors of the IJERPH.

I hope that these comments and suggestions are helpful in your work with finishing this manuscript. 

Author Response

Thank you for the opportunity to review your work. As it stands, the manuscript I very well written and deals with an important issue.

I have provided some suggestions that could help improve the manuscript e.g.

by pointing to some aspects that are unclear to an outside reader. 

Thank you very much for taking the time to read and comment on the article and for constructive help.

Abstract

In the sentence “but we do not know much about how research works with recovery-13 oriented care for the most vulnerable users with serious mental difficulty and substance use” it is not clear what is meant by “how research works with…”

We have changed the question so that it becomes clearer what we are really investigating

Introduction

  1. 1, line 32: what is meant by ‘a more humane approach’? change to holistic
  2. 1 line 35-36: I suggest rephrasing this sentence – perhaps move to method section and provide more context about the study setting. (for example, with some of the information provided in p. 2. Line 77-88) We have delete the sentence. 
  3. 1, line 37: delete ‘also’  is done
  4. 1, line 42-44: I suggest rephrasing as follows: “… in its infancy, and knowledge abouthow the concept unfolds in mental health centres, where the target group is the users with the most serious disorders, many of whom also have substance use is lackingWe have used this wording, thanks

P 2, line 45: please clarify what methods the text is referring to. we have add example 

P 2, line 67-68: I suggest rephrasing the first part of the sentence – e.g. Previous research has focused on…We have used this wording, thanks

  1. 2, line 77-82: This section seems to lack coherence – I suggest rephrasing.

Using the term ‘the special places’ makes good sense to me, as I am familiar with the Danish mental health context, however it might be difficult to follow to other readers, who are not. It could be helpful to make it very explicit, that this is actually the term used about the study setting. Perhaps simply by marking it with ‘…’ we have try to clear it more up. 

  1. 3, line 102: replace create with generate knowledge we have corected it 

Method 

This section is we written and works well to present the interview topic/question guide. I suggest providing some more demographic information about the participants such as occupation (nurses, social- and healthcare workers…), age (range or mean) and gender. Some is mentioned in the discussion/limitations, however I believe some more detailed information about this is important for readers to be able to understand the results, the quotes used etc. 

Furthermore, I suggest restructuring the section as follows – and consider changing some of the headings:

Sampling (and participants)

Focus groups

Ethical considerations

(Data) Analysis

The section on ethical considerations is repeated at the end of the manuscript – however with a different content. I suggest collating the two sections.

we have chosen to follow all these good recommendations and have corrected it in the article

Result

The results are very interesting and I get the points made. However, it would be helpful if the authors could provide some context related to the presented quotes. Also, when thinking about the study being placed within a social constructivism perspective. This could be done by providing some context about when and how the quote appeared in the focus group discussions.

Furthermore, I suggest being more explicit about the points presented as coming from the participants. As is stands, the text is not clear about this, and the lack of contextualisation around the used quotes breaks up the flow of the text.

I suggest providing a short sum-up-point at the end of the result section.

We have tried to add more information and clarify our findings

Some other specific suggestions:

The first sentence is unnecessary and can be omitted. is done

Table 2: This is perhaps a personal preference however, I would turn the table around and present the main theme then the themes and subthemes. We have discuss this but prefer the chosen structure

  1. 6, line 191: combine the sentences in line 191 and 192 with ‘and’. is corected 

Discussion

  1. 9, line 372-375: the end of this quote is not clearly marked and please add specific page number to where the quote is from. Because it is an interpretation of the overall material, we extract ambiguity in their understanding of recovery. We have tried to clarify this
  2. 9, line 387: ‘commune’ is this the right word? I suggest replacing it with municipality. agree

Conflicts of interest: Two of the authors could consider declaring that they represent guest editors of the IJERPH. We will share this information

I hope that these comments and suggestions are helpful in your work with finishing this manuscript. 

Your comments have raised the quality of the article, thanks for that

Reviewer 2 Report

Recovery-oriented practices in a mental health centre for citizens experiencing serious mental issues and substance use: As perceived by healthcare professionals

Thank you for the opportunity to review this topical and important paper, which I would like to see as published.

Review

Abstract

 Please, delete the basic titles of article sections (Intro etc.). Clarify the aims of the research and present them all over the paper consistently according to your modifications. I suggest that you give up the phrasing “how health professionals experience a recovery-oriented practice” that hints towards answers given as adverbs, like “optimistically”, “pessimistically”, “well” or “badly. Your research is rather about the concrete means of practicing recovery orientation in care work and about the elements, dimensions, outcomes, or steps of it. 

Introduction

 I would like to see here a short notion of Denmark as a Nordic welfare state that really has policies of social and health services organized, and often also financed by public national, regional or local level bodies.

 Lines 50-51: there is some confusion about the names of user movement organizations, please, correct.

 Lines 59-64: Also, there should be your own short working definition about “recovery”. On the basis of the concepts presented on these lines one wonders whether recovery is about “getting better in the sense of gaining (back) abilities to take part in social interaction/societal activities, being an active agent” or something simpler or more complicated. What is the difference between recovery-oriented practice and rehabilitation measures?

 Another possibility would be to change the aims of research, again. If there is no definition for recovery, perhaps you studied what definition can be concluded from the professionals’ experiences and about their individual or collective understanding about recovery.

 Method

 Clarifications are needed about the concrete conduct of this research as an application of qualitative content analysis.

The section “2.4. Sampling” should be presented before data and data analysis, perhaps in the connection with the section “2.1. Focus group” that seems to cover also data gathering.

The idea behind “purposive sampling” should be explained. Is it about getting relevance to the results, and at the same time about increasing validity of the research, by gathering viewpoints of various professionals as best informants, or something else? Table 1 with interview questions is about data gathering rather than about focus groups as such.

 The section “2.2. Data analysis” needs to be explained in more detail. This far the reader has understood that there is no theory to guide the analysis and starts wondering what the reasoning in extracting condensed meanings, categories, subthemes, and themes was.

Also, the reader wants to know whether the combination of all the data coming from individual participants, representing various professions in focus groups, was an adequate measure taken. It would have been adequate in cases where focus group discussions produced a common understanding. Or were there cases where the opinions of various professionals differed, according to their profession?

At least one concrete example of the process from condensed meanings to a theme would be needed here. This would also help to understand and appreciate the chosen narrative manner to report the results.

 Limitations

 Data saturation is a phase of data gathering and preliminary analysis and should thus be explained earlier in the paper.

 Other remarks

 Line 277, does “become safe” mean “feeling secure” or something else?

 Line 342, does “the massive abuse among users” mean “the massive use of drugs or alcohol” or does it mean other forms of abuse, connected to physical, psychological, or social violence, segregation etc. Line 187 seems to refer to the latter interpretation “Many of them have been abused and have themselves been abused by others” (the reader cannot see what the difference between the first and the latter parts of this sentence; they seem to describe the same phenomenon, being abused by others). Please, make this clear.

 All the best for your revision work!

Author Response

Recovery-oriented practices in a mental health centre for citizens experiencing serious mental issues and substance use: As perceived by healthcare professionals

Thank you for the opportunity to review this topical and important paper, which I would like to see as published.

Thank you very much for your help with constructive comments to improve the quality of the article

Review

Abstract

 Please, delete the basic titles of article sections (Intro etc.). Clarify the aims of the research and present them all over the paper consistently according to your modifications. I suggest that you give up the phrasing “how health professionals experience a recovery-oriented practice” that hints towards answers given as adverbs, like “optimistically”, “pessimistically”, “well” or “badly. Your research is rather about the concrete means of practicing recovery orientation in care work and about the elements, dimensions, outcomes, or steps of it. 

We have followed your recommendation, corrected and clarified

Introduction

 I would like to see here a short notion of Denmark as a Nordic welfare state that really has policies of social and health services organized, and often also financed by public national, regional or local level bodies.

We have added a text from the policy to clarify the message

 Lines 50-51: there is some confusion about the names of user movement organizations, please, correct.

We have checked up on the user movements and corrected

 Lines 59-64: Also, there should be your own short working definition about “recovery”. On the basis of the concepts presented on these lines one wonders whether recovery is about “getting better in the sense of gaining (back) abilities to take part in social interaction/societal activities, being an active agent” or something simpler or more complicated. What is the difference between recovery-oriented practice and rehabilitation measures?

 Another possibility would be to change the aims of research, again. If there is no definition for recovery, perhaps you studied what definition can be concluded from the professionals’ experiences and about their individual or collective understanding about recovery.

We have corrected the text to match the new aim and therefore target this

Method

 Clarifications are needed about the concrete conduct of this research as an application of qualitative content analysis.

We have added a more detailed explanation of how we performed content analysis

The section “2.4. Sampling” should be presented before data and data analysis, perhaps in the connection with the section “2.1. Focus group” that seems to cover also data gathering. We have follow your recommendation 

The idea behind “purposive sampling” should be explained. Is it about getting relevance to the results, and at the same time about increasing validity of the research, by gathering viewpoints of various professionals as best informants, or something else?

We have add a explanation about purposive sampling

Table 1 with interview questions is about data gathering rather than about focus groups as such. We have add it 

 The section “2.2. Data analysis” needs to be explained in more detail. This far the reader has understood that there is no theory to guide the analysis and starts wondering what the reasoning in extracting condensed meanings, categories, subthemes, and themes was.

Also, the reader wants to know whether the combination of all the data coming from individual participants, representing various professions in focus groups, was an adequate measure taken. It would have been adequate in cases where focus group discussions produced a common understanding. Or were there cases where the opinions of various professionals differed, according to their profession?

At least one concrete example of the process from condensed meanings to a theme would be needed here. This would also help to understand and appreciate the chosen narrative manner to report the results.

We have unfolded the text and tried to clarify the analytical method and show how many represent certain views, we have also added a text that explains this. We hope it meets the criticism. The idea of the case would not bring out differences in the same way as when we write, many think, some think, all think, etc.

Limitations

 Data saturation is a phase of data gathering and preliminary analysis and should thus be explained earlier in the paper. We have move it to method

 Other remarks

 Line 277, does “become safe” mean “feeling secure” or something else? corected to feeling secure

 Line 342, does “the massive abuse among users” mean “the massive use of drugs or alcohol” or does it mean other forms of abuse, connected to physical, psychological, or social violence, segregation etc. Line 187 seems to refer to the latter interpretation “Many of them have been abused and have themselves been abused by others” (the reader cannot see what the difference between the first and the latter parts of this sentence; they seem to describe the same phenomenon, being abused by others). Please, make this clear. 

We have added explanations that clarify the message

 All the best for your revision work! Thank you for helping us.